# Mother–Child and Father–Child Emotional Availability during the COVID-19 Pandemic

**DOI:** 10.3390/children10061044

**Published:** 2023-06-11

**Authors:** Maggie Dungan, Michael Lincoln, Stephen Aichele, Emma L. M. Clark, Ashley Harvey, Lillian Hoyer, Yuqin Jiao, Steffany Joslin, Frances Russell, Zeynep Biringen

**Affiliations:** Department of Human Development & Family Studies, Colorado State University, Fort Collins, CO 80528, USA

**Keywords:** COVID-19, emotional availability stressors

## Abstract

While the body of literature on COVID-19’s impacts on family life is rapidly expanding, most studies are based entirely on self-report data, leaving a critical gap in observational studies of parent–child interactions. The goal of this study was to evaluate parent–child relationships during the COVID-19 pandemic using the observational emotional availability (EA) construct. Parents (*n* = 43) were assessed using the Epidemic–Pandemic Impacts Inventory (EPII), the Flourishing Scale (FLS), and the adverse childhood experiences (ACEs) questionnaires. The subcategories of the EPII were used to develop an EPII negative and an EPII positive for each parent. EA (sensitivity, structuring, nonhostility, nonintrusiveness, child responsiveness, and child involvement) was coded from filmed parent–child interactions. Separate hierarchical multiple regressions (HMRs) were run to evaluate each of the variables of interest (EPII and FLS) as predictive of EA. Child age (*M* = 6, *SD* = 4.68) and ACEs were added in subsequent steps for EPII negative and positive if the initial step was significant. For mothers (*n* = 25), results demonstrated EPII negative as a significant predictor of EA with child age and ACEs adding only small amount of variance to the prediction. The same HMR process was repeated for flourishing, with the covariate child age alone. For fathers (*n* = 18), flourishing was a significant predictor of EA and child age added only a small amount of variance to the prediction. Results indicate that experiencing high COVID-19-related stressors is associated with lower EA for mothers, but not fathers. Having high levels of flourishing during the pandemic was predictive of higher EA for fathers, but not mothers.

## 1. Introduction

Transmission of the novel coronavirus disease (COVID-19) was declared to be at pandemic levels in March of 2020 by the World Health Organization [1]. In response to the rapid spread of COVID-19, governments worldwide implemented measures to decrease spread, by issuing stay-at-home orders and closing schools and childcare centers, thereby forcing some families to adjust to working and learning from home; others scrambling to work as essential workers, left to improvise their children’s childcare or schooling; and others finding themselves unemployed [2]. The overall impact on lives was more severe and systemic than other public health crises in the 20th and 21st centuries [3].

Emotional availability (EA) is a construct that measures the quality of the dyadic relationship between parent and child [4], and has been used in prior studies to evaluate parent–child relationships under stress such as the COVID-19 pandemic [5,6] and parents with high ACE scores [7]. The current study utilized EA to evaluate parent–child relationships in comparison to COVID-19-related outcomes (stress or flourishing).

### 1.1. Impacts of the COVID-19 Pandemic on Individuals’ Mental Health and Relationship Functioning

The COVID-19 pandemic has been reported to have had significant impacts on individuals, including a rise in mental health issues, such as overall stress, as well as depression and anxiety specifically [8,9,10]. In addition, a marked increase in intimate partner violence has been documented in the U.S. [11,12] as well as other parts of the world [13,14,15]. At more normative levels, using community samples and survey-based research, many parents reported feeling high levels of stress during the pandemic stemming from feeling trapped at home, being fully responsible for their children’s schooling or play while at home, and caring for family members if they became ill with COVID-19 [16,17]. A normative study group that began data collection from a parenting program prior to the pandemic self-reported significant negative changes in parental mental health, child behavior, and co-parenting quality from pre-pandemic levels [18]. Middle-income families self-reported declines in overall parenting quality substantially beyond reports from low- or high-income families [18].

### 1.2. Impacts of COVID-19 on Children and Families

While families’ experiences during the pandemic were diverse, preliminary research indicates psychological, economic, and social–emotional risks [19,20]. Research has shown that long-term caregivers, including parents, were significantly more likely to experience mental health challenges and fatigue than non-caregivers during the pandemic, with greater stress predicting neglectful and harsh parenting practices [17,21]. Additional research has linked the pandemic to maltreatment, especially in the context of high economic hardship, familial stress, or low social support [22].

### 1.3. Family Stress and Abusive or Neglectful Parenting Practices

The constant close contact of family members in quarantine situations at home can and did place children at increased risk of parental aggression and even violence [3,23]. Using survey data, parents who reported chronic levels of pandemic-related stress were found to be more likely to also report harsh parenting practices and engaged in abusive or neglectful behaviors in some cases [17]. Higher reported stress, anxiety, and depression were also associated with greater child abuse potential, also as reported by parents [3]. According to secondary data analysis, although child abuse and neglect increased during the pandemic, reporting and investigation dropped significantly below the levels of prior years, suggesting that most child abuse during the pandemic period went unreported [23]. This decrease in abuse reporting was likely due to a reduction in the number of mandated reporters of child neglect and abuse that children came into contact with during shelter-in-place orders and remote education [23,24].

According to self-report data, minoritized families were disproportionately impacted by COVID-19 stress [3]. The Center for Disease Control and Prevention reported that minoritized families faced an even greater risk of harmful outcomes during the pandemic as a result of systemic discrimination, a higher likelihood of living in crowded urban areas, and a greater likelihood of working essential jobs not subject to standard quarantine guidelines, with an elevated risk of contracting COVID-19 [25]. Simultaneously, severe economic hardship from pandemic-related parental job loss put these children at risk of deprivation of basic needs [3]. However, parental reports indicated that providing social support significantly helped children and families [3].

**Gender Differences**. Some studies have indicated gender differences in how parents were impacted by pandemic-related parenting challenges, with mothers continuing to bear most of the burden of childcare [8,26]. Prior research on COVID-19 and families has revealed disproportionate negative outcomes for mothers compared to fathers. During the pandemic, mothers were found to be at significant risk of developing mental health issues (internalizing problems), such as depression and anxiety, especially during pregnancy [27,28,29]. Mothers also self-reported more instances of feeling lonely and anxious, and yelling at their children, than fathers [8]. Gender differences were also found in childcare during COVID-19. According to Kerr et al. [8], 76% of mothers reported they were responsible for most of the childcare during the pandemic, in comparison to 27.8% of fathers. Relatedly, mothers who worked from home during the pandemic reported significantly higher stress than fathers who worked from home [26]. Working mothers were also more likely to reduce or eliminate their paid work hours to accommodate their children’s educational and care needs during the pandemic, while fathers did not [30]. Overall, the current survey research suggests that the pandemic may have had a significantly greater impact on mothers compared to fathers [9,18,30].

**Compensatory Role of Fathers.** Father involvement in childcare and household tasks appears to have played a role in preventing female partners from having to leave the workforce during the pandemic [30]. Additionally, fathers of young children who were more involved in childcare prior to the pandemic were significantly more likely to stop working during the initial months of the pandemic to continue helping with childcare [30]. Although women still carried a disproportionate responsibility for childcare, the pandemic may have increased the number of men serving as the primary childcare provider, indicating a redistribution of childcare in approximately a fifth of households, at least in the United Kingdom [31]. It is possible that such “forced” trends may have some societal impacts that will last well beyond the pandemic.

### 1.4. Children

Prior studies suggest that pandemic restrictions increased children’s psychological distress, including internalizing as well as externalizing problems, as reported by parents (mothers and fathers) [32]. Young children were reported to experience more significant negative impacts in these areas than older children or adolescents [32]. However, children and adolescents of varied ages (6–20 years) and around the world reported higher rates of depression, anxiety, and PTSD, which correlates with known difficulties during other major disasters, as described in the systematic review by Marques de Miranda et al. [33]. While not all children experienced major shifts in mental health symptomology, many reported behavioral changes, especially those whose parents were experiencing a greater number of COVID-19-pandemic-related stressors [16]. These behaviors included becoming fearful, anxious, or withdrawn, and increased instances of acting out [16]. Additionally, remote learning increased children’s screen time substantially, which has implications for overall health [19,34].

### 1.5. Emotional Availability

#### 1.5.1. Defining Emotional Availability

Emotional availability (EA) is defined as occurring within a dyadic relationship between caregiver and child. It is the ability of the dyad to share in a mutually beneficial, emotionally satisfying relationship, measuring the caregiver’s “receptive presence” to the child’s signals [4,35]. This receptive presence is characterized by the emotional attunement of the caregiver and the ability to respond to the information provided by their child’s emotional reactions, whether negative (e.g., distress) or positive (e.g., excitement) [36,37]. Emde [36] and Emde and Easterbrooks [37] believed that emotions serve as a barometer for the quality of a relationship. Emotional availability also utilizes a systems view towards parent–child relationships, in that individuals mutually influence and change one another through interactions [4,38]. EA consists of six dimensions: sensitivity, structuring, nonintrusiveness, nonhostility, child responsiveness, and child involvement of the caregiver. Finally, EA theory heavily relies on Bowlby’s [39] and Ainsworth and colleagues’ [40] work on attachment theory, especially in regard to the concept of caregiver sensitivity.

Adult sensitivity, the first dimension of EA, is the level to which the caregiver perceives the child’s signals and responds to them. Optimal sensitivity scores indicate high emotional warmth and attunement to the child. The second dimension of EA is structuring, or how the adult sets boundaries and creates rules for play or exploratory behaviors, while simultaneously fostering the autonomy of the child. Third, nonintrusiveness refers to the absence of “over direction, overstimulation, interference, or overprotection” [35] (p. 3), which also serves to appropriately promote autonomy while maintaining connection. The next dimension is nonhostility, referring to the absence of any hostile behaviors on the part of the adult, ranging from open to covert hostility. The last two dimensions are focused on the child’s style instead of the caregiver’s. Child responsiveness is the child’s willingness to respond to the adult’s bids for connection, supporting a balance between attachment and exploration. Child involvement of the adult is the child’s interest in including the adult in the interaction and their ability to do so effectively. These six dimensions of EA make up the EA Scales, a framework for scoring the observed parent–child interactions to evaluate levels of emotional availability [4,35]. Thus far, EA has been used for children between the ages of 0 and 14 years. This will be the first study to extend that range to 17 years, with six child participants between the ages of 13 and 17.

#### 1.5.2. Emotional Availability and COVID-19

The construct of emotional availability was utilized in one of the very few observational studies of parent–child relationships during the pandemic. In a study conducted in Israel, Shakiba and colleagues [5] used the EA Scales to assess mother–child dyads during scenarios of play and frustration, occurring before (time 1: mean infant age 3.5 months) and during the pandemic (time 2: mean infant age 12.5 months). They assessed continuity, stability, and bidirectional influences. While no significant differences in EA were found for mothers, suggestive of the normative continuity of EA over time, the children demonstrated increases in responsiveness and involvement from time 1 to time 2, which corresponds with a developmental period during which there is typically a rise in the infant’s EA [41,42]. Thus, children’s typical development of EA was not disturbed.

While a highly important study, Shakiba et al. [5] did not actually measure the COVID-19-related reactions of parents, and they also focused only on mothers. In the current study, we will measure both negative and positive experiences during COVID-19 as predictors of EA. Further, while Shakiba and colleagues [5] evaluated mother–child dyads at two time points during the pandemic, they did not include a measure to evaluate parents’ reports about their stressors or wellbeing during the pandemic. Therefore, any conclusions drawn about changes in EA during the pandemic are based on assumptions about the external context of COVID, not direct associations with measurements of feelings during the COVID-19 pandemic.

The second of very few observational studies on parent–child relationships during the pandemic also utilized emotional availability as a mechanism to improve parent–child relationships, specifically in families identified as at risk of child abuse and neglect during the COVID-19 pandemic [6]. This study found that parents and children who went through a video feedback intervention demonstrated improvements in EA compared to the control group [6]. Children in this study were between the ages of 0 and 5 and over 75% of the parents in the study were mothers. Like the study by Shakiba and colleagues [5], this study did not include a measure to evaluate parents’ levels of COVID-19-related stress specifically, so relations between EA and the pandemic are based on assumptions alone [6].

#### 1.5.3. COVID-19 Stress and Positive Change

One method to measure negative pandemic experiences was developed by Grasso and colleagues (2020a) and is known as the Epidemic–Pandemic Impacts Inventory (EPII). The EPII was created to evaluate impacts to individual and family life during the COVID-19 pandemic using a person-centered approach. Initial studies have supported the use of the EPII as a tool to evaluate both positive and negative experiences across multiple domains of life [43,44]. The initial study that utilized the EPII divided participants into distinct sociodemographic classes to compare their COVID-19 experiences across groups. Notably, individuals in one of the categories evaluated by the study “Parents—high exposure/high risk,” were more likely to hold essential jobs that required them to continue in-person work, experience childcare issues, take on teaching at home, utilize harsher discipline, report increased child behavioral problems, and were more likely to demonstrate symptoms of depression and anxiety due to high levels of stress [44]. According to this study’s assessment of various risk profiles, caregivers of children and adolescents experienced increased exposure to stress and higher psychosocial risk [44].

The EPII also surveyed positive COVID-19 experiences. Initial studies reveal that those who self-reported the most positive change during the pandemic also tended to report high levels of negative experiences on the other subcategories, indicating that the two are not mutually exclusive [44].

#### 1.5.4. Parental Wellbeing or Flourishing

Families reported very different experiences during the COVID-19 pandemic. For some families, the immense changes to daily life created opportunities for connection and increased wellbeing, reporting that the COVID-19 pandemic was a time of healing and restoration in the wake of reduced obligations [45]. Individuals who felt fulfilled in their relationships during the pandemic reported more connection with their families [19]. To better understand wellbeing during the pandemic, we drew upon the concept of “flourishing” (Diener, 2010), which refers to individuals’ feelings of success in mutually beneficial relationships and overall feelings of self-esteem, optimism, and purpose in life [46].

#### 1.5.5. Child Age

Child age has been included as a variable of interest in several self-report studies on parent–child wellbeing during the pandemic [16,32,47]. Self-report data have shown that a younger child age significantly predicted parental stress during the COVID-19 pandemic [32]. However, another self-report study found that age moderated the effect of COVID-19 on children’s emotional, cognitive, and worry reactions, such that older children were more impacted in every area [47]. Parent survey data have indicated that child age was a significant predictor of children’s acting-out behaviors, as well as anxious and withdrawn behaviors during the pandemic, such that older children demonstrated higher rates of behavioral issues [16]. Although these studies do not point to an age that is consistently “difficult”, they suggest that child age may play an important role in understanding the relationship between parent–child interactions and wellbeing during COVID-19. Research that explores the role of child age in parent–child interactions during the pandemic is both limited and conflicting, which supports the need for further evaluation of this variable.

#### 1.5.6. Adverse Childhood Experiences

As described above, parental past traumas may be a risk factor for worse child outcomes during the COVID-19 pandemic [48]. Therefore, we also wanted to ask parents about their adverse childhood experiences (ACEs). ACEs are events, such as child physical and sexual abuse, poverty, and parental substance abuse, that may alter the mental and physical health of individuals if experienced at an early age [7,49,50,51,52]. Experiencing multiple ACEs places an individual at greater risk of health and mental health problems later in life [53].

During non-pandemic times, prior research has shown that mothers who had a multitude of ACEs were more likely to report distress as well as child negative social/emotional outcomes, and that EA moderated linkages in the very-low-EA group [7]. Given we were working with a low-risk sample, we did not necessarily expect ACEs to be highly predictive of our outcomes. However, based on the prior literature on ACEs as a risk factor during times of increased stress [48], we did expect that parents’ ACE scores may be an important tool to include in our understanding of parents’ EA and COVID-19 stress.

### 1.6. The Current Study

The goal of this study was to extend the assessment model of EA to the highly unique context of the COVID-19 pandemic. The ability to compare the number of COVID-19 stressors to observational data on parent–child interactions was highly unique, as almost all studies conducted on parent–child relationships and COVID-19 so far have relied entirely on self-report data [20,54,55,56]. The exception is Shakiba et al. [5], who address the question of stability in EA from the pre-pandemic to the pandemic period.

**Hypotheses 1 (H1).** *We predicted that parents who experienced a greater volume of COVID-19-related negative stressors, and especially mothers, would demonstrate lower observed emotional availability (EA), as compared to those reporting fewer negative stressors*.

**Hypotheses 2 (H2).** *We predicted that parents who experienced a higher volume of positive changes during COVID-19 within their families would demonstrate higher EA, as compared to those reporting fewer positives. We expected that the above predictions would hold, regardless of the parents’ ACEs, in this low-risk sample*.

**Hypotheses 3 (H3).** *We predicted that parents who reported higher levels of wellbeing (i.e., flourishing) would demonstrate higher EA, as compared to those reporting lower wellbeing. Based on prior survey research, we expected mothers to be more affected by the negative stressors of the pandemic than fathers, e.g., [9,18,30], but did not have specific hypotheses related to the positives associated with the pandemic (EPII positive or flourishing)*.

**Hypotheses 4 (H4).** *We expected that all of the above predictions would hold, regardless of the child’s age, given the limited and conflicting research on the broad age range that was included in this study*.

## 2. Methods

### 2.1. Participants

The main recruitment method was via the university email distribution list of faculty staff and students. Participants were also recruited through word of mouth. The participants completed the study between August 2020 and May 2022.

Participants consisted of mothers and fathers and a child from their family, referred to as dyads (*n* = 43 total; *n* = 25 mothers; *n* = 18 fathers). Mothers and fathers were not necessarily from the same family unit. In order to be eligible, the child had to be under the age of 18 for the duration of the study. While the adult did not have to be a biological parent, they were required to be a caregiver with custody of the child.

The parents in this study were highly educated, with 70% having completed a postgraduate degree. Overall, 5% percent had completed some postgraduate education, 16% had a college degree, and 9% had completed a bachelor’s degree. Additionally, 81% percent of parent participants were married, 12% were in committed partnerships, 2% were divorced, 2% were single, and 2% were in multiple or polyamorous relationships. Parent participants ranged in age from the 25–35 category (33%) to the 45–54 category (16%), with the majority in the 35–44 category (51%).

Regarding the child participants, children in the study ranged from 4 months old to 17 years of age, with a median age of 5 and *M* = 6.25. In total, 53% of child participants were male, and 44% were female, and 2% were transgender male. Moreover, 95% of the children were biologically related to their parents and 2% were adopted (1 participant did not respond to this question).

### 2.2. Procedure

Parents who were interested in the study completed detailed consent forms approved through the university’s Internal Regulatory Board (IRB). After the completion of consent forms, participants were sent a Qualtrics link to the surveys in the study including the Epidemic–Pandemic Impacts Inventory (EPII) [43] and the Flourishing Scale (FLS) [46], followed by pretest filming for later scoring of EA. The pretest filming process required parents to engage with their child in their home in front of a camera for 20 min while they were filmed via Zoom recording. Prior to filming, parents were instructed to engage with their child in a shared activity and behave as they normally would in this interaction. After pretests, parents were randomized to an intervention vs. waitlist control group, but only time 1 (pretest data) is used in this report. As part of the intervention, some parents were asked about their positive and negative experiences during the COVID-19 pandemic. Because these questions were not asked of all parents and/or they were not asked in consistent ways, we mention some anecdotal results in the results section. Parents were sent the ACEs questionnaire via Qualtrics approximately 6 weeks after the pretest surveys due to procedural delays. All participants completed the ACEs questionnaire despite the delay.

### 2.3. Measures

#### 2.3.1. Epidemic–Pandemic Impacts Inventory (EPII)

The EPII is a 92-item questionnaire that assesses pandemic impacts on 9 dimensions of life: work, home life, social activities, economic factors, emotional health, physical health, physical distancing, infection history, and positive changes [43]. Participants were asked to respond to EPII questions based on their experiences during the height of the pandemic, regardless of when they were interviewed. For each item on the EPII, there are four response options to report the impact or relevance of the item: yes (me), yes (person in home), no, and N/A. The EPII is intended to be person-centered, allowing for self-reports of experiences or proximity to experiences (e.g., someone in the home has a pre-existing health condition that puts them at higher risk of COVID-19 complications) [43]. No psychometric properties of the EPII were available due its newness and the small number of studies that have used it thus far. However, early research demonstrates a correlation with the Perceived Stress Scale (PSS) [44,57]. The EPII demonstrated good internal reliability in the current study, with a Cronbach’s alpha of α = 0.870 for EPII negative scores, and α = 0.875 for EPII positive scores.

Creating EPII negative scores and EPII positive scores has been a common practice using this measure [44,58]. To create EPII negative scores (referred to as EPII neg), questions 1–73 of the Epidemic–Pandemic Impacts Inventory (EPII) were isolated and summed because they reference negative or stressful experiences common during the COVID-19 pandemic. Examples of such statements for the EPII neg are “hard time doing job well because of needing to take care of people in the home”, “increase in physical conflict with other adult(s) in home”, “family celebrations cancelled or restricted”, and “unable to access mental health treatment or therapy” [43]. For each participant, a cumulative negative experience score was created by summing all “yes (me)” and “yes (person in home)” responses on questions 1–73.

To create EPII positive scores (referred to as EPII pos), questions 74–92 of the EPII were isolated and summed to reflect experiences deemed as positive that individuals may have encountered during COVID-19 [43]. Examples of such statements include “more quality time with partner or spouse”, “improved relationships with family and friends”, and “increase in exercise or physical activity” [43]. A cumulative positive score was created by collapsing all “yes (me)” and “yes (person in home)” responses for questions 74–92. For the purposes of this study, all “no” and “n/a” responses were combined and scored as 0 points for EPII neg and EPII pos. This method has been used in prior studies [44,58] as a way to create cumulative scores of both negative and positive pandemic experiences.

#### 2.3.2. Flourishing Scale

The Flourishing Scale (FLS), developed by Diener et al. [46], was used to assess parental wellbeing. The 8-item Flourishing Scale reports the extent to which a person has supportive relationships, contributes to other’s happiness, experiences purpose and meaning in their life, and other markers of wellbeing. The FLS uses a Likert scale of 1–7, with 1 indicating strong disagreement and 7 indicating strong agreement with each item. The goal of the scale is to report a general score of positive functioning across various domains. Responses are added up to create a composite score, labeled “FLS Composite.” In prior studies, the FLS has demonstrated both internal validity and construct validity [46]. In the current study, the FLS composite demonstrated a Cronbach’s alpha of α = 0.90.

#### 2.3.3. Observed EA

Observed EA was evaluated using the 4th edition of the EA Scales, developed by Biringen [4,59]. There are six dimensions of observed EA, including sensitivity, structuring, nonintrusiveness, and nonhostility (parental measures), and child responsiveness and child involvement (child measures). Each dimension is assessed via a 7-point Likert scale and a 3-point Likert subscale by coders and assigned a direct score. For all scales, higher numbers reflect more desirable interactions, while lower scores indicate more problematic engagement. The meaning of the scales is shown in Figure 1.

Observed EA was coded by trained coders who had been certified reliable for the EA Scales. The EA Scales have been shown to be valid across numerous U.S. subcultures and other countries, with a test–retest reliability rate of 0.76–0.96 depending on place and time elapsed between data collection [4], as well as validity with other attachment measures (Adult Attachment Interview, Strange Situation Procedure, and Attachment Q-Sort [60].

Inter-rater reliability for mothers was established on the first 11 mother–child dyads, using intraclass correlations (ICCs), which was 44% of the total sample of mothers. The ICCs for observed EA direct scores were 0.89 for structuring, 0.79 for sensitivity, 0.78 for nonhostility, 0.58 for nonintrusiveness, 0.65 for child responsiveness, and 0.48 for child involvement. The lower reliability for nonintrusiveness was associated with having almost no variability in the sample on this variable. Child involvement also demonstrated low reliability for this sample. However, in prior research, observed EA has shown stability over time when intervention in not enacted [4].

Inter-rater reliability for fathers was established in the following way: Two trained coders scored videos and scores were compared to establish inter-rater reliability for fathers. The six observed EA dimensions were coded for the first 10 father–child dyads, which was 56% of the total sample of fathers. The Cronbach’s α for observed EA direct scores was 0.71 for structuring, 0.77 for sensitivity, 0.79 for nonhostility, 0.99 for nonintrusiveness, 0.85 for child responsiveness, and 0.85 for child involvement.

#### 2.3.4. Adverse Childhood Experiences (ACEs)

The Adverse Childhood Experience (ACE) [52] questionnaire was administered to both mothers and fathers, but approximately 6 weeks after the initial surveys due to procedural delays in adding this measure. ACE scores are determined based on level of childhood exposure to various potentially harmful circumstances, such as poverty, abuse, violence, and divorce. For the purpose of this study, the ACE questions were posed to the parent to assess the number of potentially traumatic experiences they encountered before the age of 18. According to prior research, ACEs demonstrate stability over time [7], as the measure assesses if the respondent has experienced any of the items prior to the age of 18, and reporting does not appear to change based on timing of assessment in adulthood. This score provides a picture of the pre-existing states of parents not related to living during the COVID-19 pandemic. Previous studies have found that the retrospective assessment of ACEs has good to excellent test–retest reliability over time, and various measures of ACEs are shown to be highly interrelated and correlated. The ACEs questionnaire demonstrated a Cronbach’s alpha of α = 0.791 in the current study.

### 2.4. Plan of Analytic Procedures

Hierarchical multiple regression analyses (HMRs) were used to test the relation between variables of interest (EPII neg, EPII pos, FLS, ACEs, and child age) and observed emotional availability (EA) dimensions. This analysis was chosen because we wanted to add variables to the model in separate steps, with the goal of controlling for specific covariates. The main goal of this study was to assess the relationship between COVID-19 stress (EPII negative score) or resilience (EPII positive score) and EA. The HMR approach allowed us to enter the main variable of interest and then funnel down to covariates if there was a significant relation, while prioritizing the amount of variability in the outcome variable (EA dimensions) that was caused by the predictor variable (EPII or FLS). Unless there was no relation between EPII neg (or EPII pos) and the dependent variable, we then used child’s age and ACEs as covariates for the HMRs testing of EPII neg and pos. Child age was evaluated as a covariate, since child participants ranged from 4 months to 17 years of age, to control for effects related to developmental level. ACEs was used to measure the level of stress and/or trauma in parents prior to the pandemic period. An additional goal was to evaluate the role of parental wellbeing during the pandemic (measured via the Flourishing Scale). Flourishing was utilized to capture a measure of parents’ psychological resources during the pandemic period. Here, as well, unless there was no relation between flourishing and the dependent variable, child age was included as a covariate for flourishing to control for developmental-based influences on parent–child relationships.

Each outcome measure was initially tested for skewness and kurtosis in SPSS, with absolute values > 2.0 for either measure considered as evidence of meaningful departures from normality. Three sets of regressions for each parent gender (mothers and fathers) were then conducted to evaluate the relation between EPII neg and EA, EPII pos and EA, and flourishing and EA. These variables were entered as step one for each of the six dimensions of EA. The main variables of interest (EPII neg, EPII pos, and FLS) were evaluated by separate HMRs at step one in order to evaluate the direct relationship between EA and the main variables, before testing for the amount of variance added by covariates. If the initial step was significant, covariates were added (age of child and ACEs for the HMRs conducted on the prediction of EA from EPII scores; age of child for the HMRs conducted on the prediction of EA from flourishing) to evaluate if the relation remained significant after controlling for other factors. Child age was included at the second step for EPII neg, EPII pos, and flourishing in order to control for developmental effects for significant relations. Parent ACE scores were only included at the third step with child age for EPII neg and EPII pos, due to the literature on the relation between prior trauma and COVID-19 stress [48]. ACEs were not theoretically linked to flourishing for the purposes of this study, so were not included in the flourishing HMRs.

## 3. Results

### 3.1. Preliminary Analyses

The zero-order correlations for the mothers indicate significant correlations as follows: EPII neg for sensitivity (r = −0.51, *p* < 0.01), nonhostility (r = −0.45, *p* < 0.05), child responsiveness (r = −0.48, *p* < 0.05), and child involvement (r = −0.44, *p* < 0.05). There are no significant correlations between EPII neg and EA variables for the fathers. Additionally, there are no significant correlations for mothers or fathers between the EPII pos and EA variables. On the other hand, for fathers, FLS is significant with sensitivity (r = 0.53, *p* < 0.05), structuring (r = 0.55, *p* < 0.05), and child responsiveness (r = 0.56, *p* > 0.05). For mothers or fathers, there are no significant correlations between ACEs and EA or between EPII neg and EPII pos.

### 3.2. Hierarchical Multiple Regressions (HMRs)

#### 3.2.1. EPII Negative Score (EPII Neg)

**Mothers.** EPII neg significantly predicted maternal sensitivity (β = −0.51, *p* ≤ 0.01), and explained 26% of the variance (F (1, 22) = 7.84, *p* < 0.01), indicating that higher EPII neg scores predicted lower observed sensitivity in the mother. At step 2, child age predicted an additional 2.6% of the variance in maternal sensitivity (F (2, 21) = 0.77, *p* = 0.39). At step 3, child age and ACEs together predicted an additional 3.2% of the variance in maternal sensitivity (F (3, 20) = 0.95, *p* = 0.34), showing that child age and ACEs did not add much to the prediction of maternal sensitivity over and above EPII neg.

EPII neg significantly predicted maternal nonhostility (β = −0.46, *p* < 0.05), and explained 21% of the variance (F (1, 22) = 5.81, *p* < 0.05), indicating that higher EPII neg scores predicted lower observed nonhostility in the mother. At step 2, child age predicted an additional 7.4% of the variance in maternal nonhostility (F (2, 21) = 2.17, *p* = 0.16). At step 3, child age and ACEs together predicted an additional 5.8% of the variance in maternal nonhostility (F (3, 20) = 1.76, *p* = 0.20), showing that child age and ACEs did not add a great deal to the prediction of maternal nonhostility over and above EPII neg.

EPII neg significantly predicted child responsiveness to mothers (β = −0.48, *p* < 0.05), and explained 23% of the variance (F (1, 22) = 6.70, *p* < 0.05), indicating that higher EPII neg scores predicted lower observed child responsiveness in the children of mothers. At step 2, child age predicted an additional 3.5% of the variance in maternal child responsiveness (F (2, 21) = 0.995, *p* = 0.33). At step 3, child age and ACEs together predicted an additional 2.2% of the variance in maternal child responsiveness (F (3, 20) = 0.632, *p* = 0.436), showing that child age did not add a great deal to the prediction of maternal child responsiveness over and above EPII neg.

EPII neg significantly predicted child involvement (β = −0.45, *p* < 0.05), and explained 20% of the variance (F (1, 22) = 5.51, *p* < 0.05), indicating that higher EPII neg scores predicted lower observed child involvement in the children of mothers. At step 2, child age predicted an additional 1% of the variance in maternal child involvement (F (2, 21) = 0.134, *p* = 0.718). At step 3, child age and ACEs together predicted an additional 0.4% of the variance in maternal child involvement (F (3, 20) = 0.009, *p* = 0.757), showing that child age contributed only minimally to the prediction.

The bivariate EPII negative and EA associations were not significant for maternal structuring or maternal nonintrusiveness. Therefore, covariates were not added in subsequent steps and these two variables were not evaluated further (Table 1).

**Fathers.** The bivariate EPII negative and EA associations were not significant for any of the dimensions of EA for fathers. Therefore, covariates were not added in subsequent steps and this variable was not evaluated further for fathers (Table 1).

#### 3.2.2. EPII Positive Score (EPII Pos)

**Mothers and Fathers.** As shown in Table 1, the bivariate EPII positive and EA associations were not significant for any of the dimensions of EA for mothers or fathers. Therefore, covariates were not added in subsequent steps and this variable was not evaluated further (Table 1).

#### 3.2.3. Flourishing

**Mothers.** One bivariate association demonstrated a relation at a trend level (*p* = 0.062) for mothers (child responsiveness), so the decision was made to not proceed any further with testing covariates (Table 1).

The other bivariate FLS and EA associations were not significant for maternal sensitivity, nonhostility, structuring, nonintrusiveness, and child involvement. Therefore, covariates were not added in subsequent steps and these five variables were not evaluated further.

**Fathers.** Hierarchical multiple regressions (HMRs) were used, entering FLS at the first step and child age at the second step. Each dimension of EA was entered as the outcome variable in independent HMRs (Table 1).

FLS significantly predicted paternal sensitivity (β = 0.531, *p* < 0.05), and explained 28% of the variance (F (1, 16) = 6.27, *p* < 0.05), indicating that higher FLS predicted higher observed sensitivity in fathers. Child age contributed 0% variance to the prediction, which is not significant (F (2, 15) = 0.006, *p* = 0.938).

FLS significantly predicted paternal child responsiveness (β = 0.561, *p* < 0.05), and explained 32% of the variance (F (1, 16) = 7.35, *p* < 0.05), indicating that higher FLS predicted higher observed child responsiveness in the children of fathers (β = 0.548, *p* < 0.05). Child age accounted for an additional 0.3% of the variance in paternal child responsiveness (F (2, 15) = 0.057, *p* = 0.815).

FLS significantly predicted paternal child involvement (β = 0.467, *p* = 0.05), and explained 22% of the variance (F (1, 16) = 4.47, *p* = 0.05), indicating that higher FLS predicted higher observed child involvement in the children of fathers. Child age accounted for an additional 5% of the variance in paternal child involvement (F (2, 15) = 1.08, *p* = 0.315).

FLS significantly predicted paternal structuring (β = 0.554, *p* < 0.05), and explained 31% of the variance (F (1, 16) = 7.07, *p* < 0.05), indicating that higher FLS predicted higher observed structuring in the children of fathers. Child age accounted for an additional 1% of the variance in paternal structuring (F (2, 15) = 0.250, *p* = 0.062).

The bivariate FLS and EA associations were not significant for paternal nonhostility or paternal nonintrusiveness. Therefore, covariates were not added in subsequent steps and these three variables were not evaluated further (Table 1).

## 4. Discussion

The purpose of this study was to better understand parent–child relationships in the unique context of the COVID-19 pandemic, and as such it is the first study to study how COVID-19-related stressors (negative and positive) as well as overall wellbeing (i.e., flourishing) predict observed parent–child relationships. One prior study [5] conducted observations during the pandemic and used EA, but their research questions involved stability in EA from pre-pandemic to pandemic levels, in contrast to studying the impact of the pandemic on families.

First, we predicted that mothers and fathers who experienced more negative stressors during the COVID-19 pandemic would show less EA, as compared to those reporting fewer negative stressors. Second, we predicted that mothers and fathers who experienced positive events during the COVID-19 pandemic would show higher EA, as compared to those reporting fewer positives. We expected that these predictions would hold regardless of parent ACEs in this low-risk sample. Third, we predicted that mothers and fathers who reported higher levels of flourishing would show higher EA, as compared to those reporting lower flourishing. Based on prior survey data, we expected mothers to be more affected by COVID-19 negative stressors than fathers, e.g., [9,18,30], but did not postulate specific hypotheses related to the positives associated with the pandemic (EPII positive or flourishing). Finally, we expected that the above predictions would hold, regardless of the child’s age. Briefly, our findings indicate meaningful relations between EPII neg and EA for mothers, no significant relations between EPII pos and EA for mothers or fathers, and significant relations between flourishing and EA for fathers during the pandemic period.

### 4.1. Mothers

#### 4.1.1. EPII Negative

Hierarchical multiple regressions (HMRs) indicated that EPII negative stressors predicted observed EA (sensitivity, nonhostility, child responsiveness, and child involvement). These findings indicate that experiencing higher levels of COVID-19-related negative stressors lead to mothers having lower observed relationship quality with their children. Significant findings for two of the parental dimensions (sensitivity and nonhostility) as well as both child dimensions (child responsiveness and child involvement) at step 1 indicate that maternal COVID-19 stress significantly impacted both mother and child. This finding is somewhat surprising, considering EPII neg evaluated the stress of the parent, not the child. However, this finding aligns with the dyadic nature of EA in which parent and child mutually influence and respond to one another [4], indicating children were sensitive to their mother’s stress and reduced their responsiveness and involving behaviors accordingly.

This finding is an important contribution to the field of family research during COVID-19, as it shows the interaction of parental stress on the dyadic parent–child relationship in observed interactions. Other research on parent–child relationships during COVID-19 focused heavily on populations with high rates of child abuse and neglect and relied almost entirely on self-report data [3,19,20,29,61]. In this normative and highly resourced sample, the relationship between EA and EPII neg indicates that mothers and their children mutually affected one another’s behavior in nuanced ways. Interestingly, prior self-report studies have suggested that maternal hostility (not measured using EA) increased during the pandemic [62,63]. Our finding that increases in EPII neg lead to decreases in sensitivity and nonhostility not only align with these findings but provide a direct comparison between observational data and individual reported experiences of COVID-19-related stress.

#### 4.1.2. Child Age and ACES

These two variables were entered at the second step for the EPII variables to see if they added additional variance to the prediction of EA, and, generally speaking, they did not. In relation to child age, this finding means that the patterns observed were present for moms regardless of the age of their child. This is substantial because mothers in this study group had children ranging in age from infancy to 17, a range rarely seen in observational parent–child intervention studies. Particularly in the context of the COVID-19 era, this finding is significant. Impacts on mother-infant relationships during COVID-19 have been well documented [27,29,61]. These mothers may have felt solely responsible for the care of their child in the face of isolation, as well as increased fears about infection for themselves and their babies who could be vaccinated or masked easily. However, mothers of elementary, middle, and high schoolers had the additional support of teachers, albeit remotely, in the education and support of their children. Despite these differences, our findings suggest that impacts to EA were ubiquitous across child development during the pandemic.

These findings also indicate that maternal ACEs did not predict EA over and above EPII neg and child age, meaning mothers were impacted similarly regardless of their trauma history. This finding is surprising due to the existing literature suggesting ACEs to be a risk factor for negative outcomes during the pandemic [64,65], but may be due in part to the limited size and diversity of the study population. The findings are also interesting because the mothers varied between 0 and 11 out of 16 items, and hence showed variability in their responses. Regardless, this was a low-risk group of mothers.

#### 4.1.3. EPII Positive

Surprisingly, no significant relations between EPII pos and EA were found for mothers. This finding may indicate positive changes during COVID-19 that worked to counteract stressors but did not result in meaningful improvements in mother–child interactions. Alternatively, COVID-19 stressors may have outweighed the impacts of positive experiences during the pandemic, resulting in the findings above.

#### 4.1.4. Flourishing

There were no significant findings that reached statistical significance for mothers. For this sample, flourishing was not a meaningful tool by which to evaluate mother–child EA during the pandemic. This finding is surprising in comparison to the outcomes for fathers, detailed below.

### 4.2. Fathers

#### 4.2.1. EPII Negative

No significant relations were found between EPII neg and any dimensions of EA for fathers. This surprising finding indicates that COVID-19 stress did not meaningfully impact fathers’ parenting behavior in this sample, or we were not able to see it due to small sample size. If real, however, it may be due to cultural or traditional views of fathers as the secondary caregiver, who steps in when needed but is not the primary respondent to child needs. During the shift to working at home, fathers may have been less likely to experience role strain related to childcare because they relied on their female partner to be the default parent, even if she was also maintaining a job from home.

#### 4.2.2. EPII Positive

Like mothers, no significant relationships between EA and EPII pos were found for fathers. Similarly, this finding may indicate that positive experiences during COVID-19 were not enough to outweigh COVID-19 stressors or did not meaningfully improve fathers’ parenting behaviors.

#### 4.2.3. Flourishing

Significant relations were found between flourishing and four dimensions of EA for fathers: sensitivity, structuring, child responsiveness, and child involvement. This finding indicates that fathers who self-reported higher psychological wellbeing also demonstrated higher EA. The significant findings for both of the child dimensions suggest that children were meaningfully impacted by the positive state of their fathers. Different from mothers, structuring and flourishing were positively related for fathers, showing that flourishing directly impacted paternal behavior, in addition to child behaviors. While these findings with flourishing were not directly compared to COVID-19 variables, the entirety of this study occurred during the pandemic period. This finding further supports that flourishing may serve as a protective factor against negative outcomes related to external stress, such as the COVID-19 pandemic.

#### 4.2.4. Child Age

Our findings demonstrate that child age did not predict EA over and above flourishing for fathers. Similar to moms, this finding is significant due to the lack of observational parent–child intervention studies that cover multiple developmental levels of children. This research is even more lacking for fathers than mothers. Children of father participants ranged in age from infancy to 14 years old, reflecting a wide array of developmental levels. Regardless of the age of the child, fathers who self-reported high flourishing also demonstrated high EA (sensitivity, structuring, and child responsiveness) during observed interactions. This finding means that work to support the flourishing of fathers has the potential to positively impact their relationships with their children, no matter their age.

### 4.3. Parent Gender

Running separate HMR tests for parent–child dyads by parent gender allowed us to evaluate the unique interactions of COVID-19 stress and wellbeing and the six EA dimensions for mothers and fathers independently. Our findings suggest that mother–child EA was significantly impacted in relation to COVID-19 stress, while fathers’ EA was not. Additionally, our findings suggest that experiencing wellbeing, or flourishing, during the pandemic period, was more impactful on fathers’ EA, than mothers’. These outcomes may indicate that mothers carried more of the household stress during the pandemic, leading to decreases in the quality of their parenting. Fathers, on the other hand, were not as impacted. Fathers who were doing particularly well and reported high flourishing were parenting in a way that related to higher-quality parent–child relationships, despite the external context of the pandemic. Prior self-report studies have found evidence that the social context of COVID-19 may have led to fathers taking a more active role in parenting. Simultaneously, many mothers experienced further inequity, reducing their paid work hours and still managing a much higher percentage of household tasks and childcare than their male partners [31]. In this context, it makes sense that mothers were more impacted by levels of COVID-19-related stress than fathers, as mothers were still the primary managers of their households, even if their male partners provided increased assistance compared to pre-pandemic levels.

### 4.4. COVID-19 Experience Interviews

The findings from this study were extracted from a larger study, which included an intervention that involved some interview questions. During the one-on-one coaching sessions with each parent participant, the EA coach posed questions about COVID-19 experiences and parenting. These questions included “how have your interactions changed with your child since the start of the pandemic”, “what does a typical day look like for you”, “have you had to make adjustments to work from home”, “what concerns do you have regarding your experience parenting during COVID”, and “what positive changes have you noticed during this transition—have you seen any aspects of your relationship dynamics with your child strengthened?” These questions were intended to capture anecdotal evidence of parents’ experiences of parenting during the pandemic, allowing for further contextualization of their EA scores and responses to the intervention. These questions were posed in the conversational manner that was used throughout the parent coaching, so responses were too variable to code or incorporate in statistical analysis. Instead, the parents who reported high for each EPII score (negative and positive) and low for each EPII score (negative and positive) were selected, and their COVID-19 interviews were combed to provide anecdotal context for their scores (Table 2). A common theme across all parent interviews, regardless of EPII score, was a combination of positive and negative experiences. Anecdotally, it appears that families encountered a unique combination of challenges and opportunities for positive change during the pandemic that may have an important influence on their relationships going forward. Parents’ stories of positive change, especially related to connection with their children, indicate resiliency effects which may be an important are of inquiry for future family research.

### 4.5. Strengths, Limitations, and Future Directions

While recruitment strategies were persistent and diversified, recruiting participants, especially fathers, was challenging, hence the small sample size. Additionally, several attempts were made to connect with local organizations that serve higher-risk populations to recruit a more diverse sample, but these efforts did not result in successful recruitment of participants. Future research would benefit from evaluating a more socioeconomically and ethnically diverse sample. Additionally, the small sample sizes for each parent gender (mothers *n* = 25, fathers *n* = 18) should be considered when evaluating the generalizability of the findings of this study. Finally, data for this study were collected over an extended period of time, from August 2020 to May 2022. Participant responses to questionnaires and their observed EA may have been impacted by timing, due to the changes in daily life as a result of differing levels of COVID-19 restrictions at various time points. There was not an adequate way to assess the variable of time within this study, so this must be considered as a limitation when evaluating differences in the parent–child relationships.

We also considered testing child age as a covariate following categorization based on developmental window (e.g., younger vs. older children). However, we are cognizant that the dichotomization of a continuous variable almost always results in information loss (reduction in individual variability). Because our sample was already small, and because children were not inducted into the study based on age cohorts, we chose to preserve age as-is. It will be important in future studies with larger samples to more fully consider developmental differences (including explanatory variables in addition to age cohort). The current study evaluated mothers and fathers from independent families, instead of evaluating couples who parent together. This model allowed for the evaluation of a larger community sample, analyzing the experiences of over 40 different households. However, since the parent participants were not paired, conclusions about the interplay between mothers’ and fathers’ experiences, or their dyadic interactions, cannot be drawn from this study. The potential compensatory role of fathers as the “secondary caregiver” cannot be evaluated, as we do not know how the partners of the fathers in this study would have responded differently regarding COVID-19 stress or observed EA. Furthermore, we did not collect data on the quantity of parenting time for each parent, either solo or co-parenting time.

The Epidemic–Pandemic Impacts Inventory (EPII), used to measure COVID-19 stress in this study, is a fairly new measure that is yet to undergo rigorous validity and reliability testing. It is yet to be determined if this measure, or a different one, is the most accurate evaluation tool for pandemic-related stress. It is yet to be seen if this measurement tool will be evaluated further, as collecting data on COVID-19-related experiences will become more challenging as the pandemic moves further into the rearview for society.

## Figures and Tables

**Figure 1 children-10-01044-f001:**
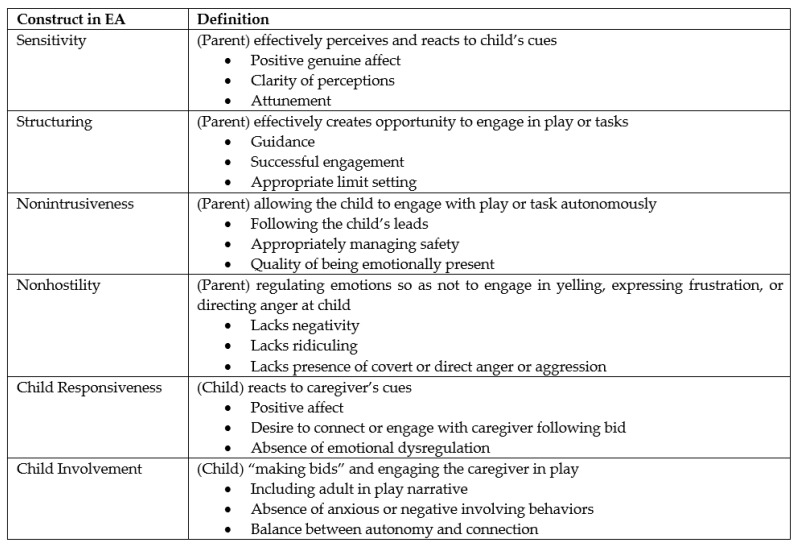
Definitions of Constructs used in EA Scales.

**Table 1 children-10-01044-t001:** Effect of EPII and FLS on EA dimensions.

Outcome ^a^	Covariates ^b^	Moms			Dads		
		Rsq	Beta	*p*	Rsq	Beta	*p*
		**Predictor = EPII Negative (EPII neg)**					
Sensitivity	EPII neg	0.263	−0.513	0.010	0.018	0.136	0.591
	+Age	0.289	−0.464	0.025			
	+Age & ACEs	0.321	−0.384	0.081			
Nonhostility	EPII neg	0.209	−0.457	0.025	0.014	0.119	0.638
	+Age	0.283	−0.375	0.065			
	+Age & ACEs	0.341	−0.269	0.207			
Child Resp.	EPII neg	0.234	−0.483	0.017	0.065	0.254	0.309
	+Age	0.268	−0.427	0.040			
	+Age & ACEs	0.291	−0.361	0.107			
Child Invo.	EPII neg	0.200	−0.448	0.028	0.017	0.129	0.609
	+Age	0.205	−0.426	0.048			
	+Age & ACEs	0.209	−0.399	0.093			
Structuring	EPII neg	0.134	−0.366	0.079	0.052	−0.228	0.362
Nonintrusiveness	EPII neg	0.007	0.083	0.701	<0.001	0.001	0.997
		**Predictor = EPII Positive (EPII pos)**					
Sensitivity	EPII pos	0.075	0.274	0.186	0.022	0.150	0.554
Nonhostility	EPII pos	0.017	0.131	0.532	0.048	−0.218	0.384
Child Resp.	EPII pos	0.021	0.144	0.493	0.089	0.299	0.228
Child Invo.	EPII pos	0.015	0.123	0.559	0.128	0.358	0.145
Structuring	EPII pos	0.020	0.140	0.504	0.004	0.063	0.804
Nonintrusiveness	EPII pos	0.049	−0.222	0.287	0.025	0.158	0.531
		**Predictor = Flourishing (FLS)**					
Sensitivity	FLS	0.086	0.293	0.155	0.282	0.531	0.023
	+Age				0.282	0.535	0.032
Nonhostility	FLS		0.214	0.304	0.086	0.293	0.237
Child Resp	FLS	0.143	0.379	0.062	0.315	0.561	0.015
	+Age				0.317	0.548	0.025
Child Invol.	FLS	0.079	0.280	0.174	0.218	0.467	0.050
	+Age				0.271	0.410	0.091
Structuring	FLS	0.004	−0.062	0.770	0.306	0.554	0.017
	+Age				0.318	0.580	0.019
Nonintrusiveness	FLS	0.001	−0.038	0.857	0.193	0.439	0.068

^a^ Across outcomes, SPSS skewness ranged from −1.55 to 0.10 and kurtosis ranged from −1.80 to 1.10 and were thus within the ±2.00 range that is considered acceptable. ^b^ Covariates were examined only when the bivariate EPII-EA association or the FLS-EA association was significant.

**Table 2 children-10-01044-t002:** Anecdotes from Parent Interviews on COVID-19.

Moms		Comments
	High EPII Negative	Anger from teenager about isolation, increase in flexibility and adaptability in children, separation from husband during COVID-19 lead to housing challenges, extreme stress for children during housing instability and separation, shift to working full time during COVID-19 to provide for family, older child took care of younger child during some of at-home schooling, increased closeness between mother and children.
	Low EPII Negative	Family spent more time together, more time outside, more quality time as a family; working from home improved parenting due to availability; working from home created unique challenges, difficult to work with children around and to wait for husband to respond to children; increased fear of children getting sick or having to go into NICU.
	High EPII Positive	Worked from home prior to pandemic so little changed; child liked being home a lot initially and did not feel overly isolated, decided to keep child at home during first phase of going back to school which lead challenges with child’s education; mother liked being pregnant during the pandemic so she could work from home and receive support from nearby family, older child got to help take care of infant child.
	Low EPII Positive	Increased responsibility to manage child’s learning, more time together as a family, increased concern for extended family member’s health, child stepped up to help with elderly family members.
**Dads**		
	High EPII Negative	Wife worked in healthcare and increased work and income during COVID-19, balancing children’s schooling and his work was challenging; dealing with overwhelm, frustration, and more conflict with children; balancing multiple roles (teacher, dad, faculty); oldest child’s motivation for school decreased severely; father felt he entered survival state for a long period of time; increased time together during breaks from school and work; increased time to bond with children and engage in collaborative play; more time outside and less stress attending multiple classes and events.
	Low EPII Negative	Infant has more interactions over Zoom with extended family than in person but recognizes and reaches for them. Cannot get together with some family because of vaccination status and illness, concerned child may not recognize family members who were wearing masks. Father experienced minimal shifts in work and enjoyed more time parenting, able to reduce hours child spent in childcare.
	High EPII Positive	Prior homeschooling experience made transition easier; father gave up time working to parent, took over all meal and morning/night routines while wife worked, found time to engage in hobbies, did not have financial difficulties because wife worked in healthcare; family lived in the mountains and were naturally isolated and close to outdoors; older child appreciated regular school more after schooling from home; children’s ability to self-manage time improved
	Low EPII Positive	Challenging to share split custody due to travel restrictions—limited father’s ability to see child; worried about lack of social opportunities and peer learning opportunities for children; concerned that most of child’s time was spent alone with mother.

## Data Availability

Data will be publicly available, except for videotapes.

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
