# Peer review of "Mother–Child and Father–Child Emotional Availability during the COVID-19 Pandemic"

_children, 2023, doi:10.3390/children10061044_

Round 1
Reviewer 1 Report
I read the article carefully, I believe it is an interesting and important article. I propose to make changes mainly in the literature review chapter which seems to me to be too long. I recommend introducing the concept of emotional availability at an earlier stage of the review, linking it to the corona epidemic and the challenges it posed, and then introducing the aim of the current study. The reader should know from the beginning what the aim is.
In addition, I would like to know whether age as a predictor variable was examined in the statistical analysis as a ‘dummy variable’ of young age versus old age or in some other way. If this is not done it is recommended to check it this way and then the results may be different
Author Response
I read the article carefully, I believe it is an interesting and important article. I propose to make changes mainly in the literature review chapter which seems to me to be too long. I recommend introducing the concept of emotional availability at an earlier stage of the review, linking it to the corona epidemic and the challenges it posed, and then introducing the aim of the current study. The reader should know from the beginning what the aim is.
We have modified the introduction to foreshadow the key questions of this study and highlight this in the introduction.
In addition, I would like to know whether age as a predictor variable was examined in the statistical analysis as a ‘dummy variable’ of young age versus old age or in some other way. If this is not done it is recommended to check it this way and then the results may be different.
Thank you for noting this. We have added the following text as a study limitation (along with our explanation for not choosing to dichotomize child age as a covariate).
We also considered testing child age as a covariate following categorization based on developmental window (e.g., younger vs. older children). However, we are cognizant that dichotomization of a continuous variable almost always results in information loss (reduction of individual variability). Because our sample was already small, and because children were not inducted into the study based on age cohorts, we chose to preserve age as-is. It will be important in future studies with larger samples to more fully consider developmental differences (including explanatory variables in addition to age cohort).
Reviewer 2 Report
Thank you for the opportunity to review this study entitled “Mother-Child and Father-Child Emotional Availability During the COVID-19 Pandemic” (children-2430275).
The research focused on the parent-child relationships, in the context of the the COVID-19 pandemic. A sample of 43 parents (n = 25 mothers; n = 18 fathers) have been involved in the research.
In my opinion, the research topic is relevant, and the study is interesting. Parallelly, there are some issues that need to be addressed.
· Abstract: the information about the sample should be deepened (Mean age and SD?) to provide a clear picture of what will be presented in the paper.
· Introduction: in this section, it would be useful to organize the subparagraphs. A first heading “1. Introduction” should be added in the initial phase of the text. After that, one should proceed with subparagraphs: “1.1 Impacts of the COVID-19 Pandemic on Individuals’ Mental Health and Relationship Functioning”; “1.2 Impacts of COVID-19 on Children and Families” etc…
· “participants completed the study between February 2020 and March2022.” It is a very long period and evaluating the psychological consequences of the COVID-19 in subjects recruited in such a wide range also implies an additional intervening variable: the time elapsed since the beginning of the pandemic. This aspect is a important limitation of the study, and should be discussed.
· The limit of the small sample size should be further stressed since this implies the need to be cautious when generalizing the results of this study.
Best wishes
Author Response
Abstract: the information about the sample should be deepened (Mean age and SD?) to provide a clear picture of what will be presented in the paper.
This has been added and highlighted in yellow.
Introduction: in this section, it would be useful to organize the subparagraphs. A first heading “1. Introduction” should be added in the initial phase of the text. After that, one should proceed with subparagraphs: “1.1 Impacts of the COVID-19 Pandemic on Individuals’ Mental Health and Relationship Functioning”; “1.2 Impacts of COVID-19 on Children and Families” etc…
Thank you for this suggestion. This has been done now.
“participants completed the study between February 2020 and March2022.” It is a very long period and evaluating the psychological consequences of the COVID-19 in subjects recruited in such a wide range also implies an additional intervening variable: the time elapsed since the beginning of the pandemic. This aspect is a important limitation of the study, and should be discussed.
This has now been addressed in the discussion section as a limitation and highlighted in yellow.
The limit of the small sample size should be further stressed since this implies the need to be cautious when generalizing the results of this study.
Thank you, and this also is additionally noted in the discussion section.